 **eLIFE**

# Fatty acid analogue N-arachidonoyl taurine restores function of I$_{Ks}$ channels with diverse long QT mutations

Sara I Liin[1,2]*, Johan E Larsson[2], Rene Barro-Soria[1], Bo Hjorth Bentzen[3,4], H Peter Larsson[1]*

[1]Department of Physiology and Biophysics, University of Miami, Miami, United States; [2]Department of Clinical and Experimental Medicine, Linköping University, Linköping, Sweden; [3]The Danish Arrhythmia Research Centre, University of Copenhagen, Copenhagen, Denmark; [4]Department of Biomedical Sciences, University of Copenhagen, Copenhagen, Denmark

**Abstract** About 300 loss-of-function mutations in the I$_{Ks}$ channel have been identified in patients with Long QT syndrome and cardiac arrhythmia. How specific mutations cause arrhythmia is largely unknown and there are no approved I$_{Ks}$ channel activators for treatment of these arrhythmias. We find that several Long QT syndrome-associated I$_{Ks}$ channel mutations shift channel voltage dependence and accelerate channel closing. Voltage-clamp fluorometry experiments and kinetic modeling suggest that similar mutation-induced alterations in I$_{Ks}$ channel currents may be caused by different molecular mechanisms. Finally, we find that the fatty acid analogue N-arachidonoyl taurine restores channel gating of many different mutant channels, even though the mutations are in different domains of the I$_{Ks}$ channel and affect the channel by different molecular mechanisms. N-arachidonoyl taurine is therefore an interesting prototype compound that may inspire development of future I$_{Ks}$ channel activators to treat Long QT syndrome caused by diverse I$_{Ks}$ channel mutations.

*For correspondence: sara.liin@ liu.se (SIL); PLarsson@med.miami. edu (HPL)

## Introduction

Long QT syndrome (LQTS) is a condition of the heart which in most cases is caused by a mutation in cardiac ion channels (*Hedley et al., 2009*; *Morita et al., 2008*). In LQTS, the action potential of the heart is prolonged, which is observed as a prolonged QT interval in the electrocardiogram. LQTS patients have an increased risk of developing ventricular tachyarrhythmias called *torsades de pointes* when exposed to triggers such as adrenergic stress (*Morita et al., 2008*; *Cerrone et al., 2012*). These arrhythmias can cause palpitation, syncope or sudden death due to ventricular fibrillation. To improve the clinical outcome of LQTS patients, it is therefore critical to prevent these LQTS-induced life-threatening arrhythmias.

Most mutations causing LQTS are located in the *KCNQ1* gene (*Hedley et al., 2009*). *KCNQ1* codes for the potassium channel K$_V$7.1, which in the heart co-assembles with the beta-subunit KCNE1 to form the slowly-activating, voltage-dependent potassium channel I$_{Ks}$ (*Barhanin et al., 1996*; *Sanguinetti et al., 1996*). The I$_{Ks}$ channel provides one of the important delayed rectifier outward potassium currents that repolarizes the cardiomyocyte and terminates the cardiac action potential (*Nerbonne and Kass, 2005*). Reduced I$_{Ks}$ function therefore tends to delay cardiomyocyte repolarization, thereby causing prolonged cardiac action potential durations and a prolonged QT interval. The cardiac I$_{Ks}$ channel consists of four K$_V$7.1 subunits and two to four KCNE1 subunits (*Nakajo et al., 2010*; *Plant et al., 2014*; *Murray et al., 2016*). Throughout this work, we will refer to

**eLife digest** Every heartbeat relies on an electric wave that travels through the heart. This wave must reach different parts of the heart in a specific sequence to ensure that the heart muscle cells contract in a coordinated manner. Such coordinated contractions enable the heart to pump enough blood around the body. By allowing specific ions to flow into or out of the heart muscle cell, proteins called ion channels in the cell membrane generate the electric wave, keep it going and stop it. One such protein called the $I_{Ks}$ channel controls the flow of potassium ions, and in doing so stops the electric wave in heart muscle cells.

About 300 different mutations in the $I_{Ks}$ channel have been shown to cause abnormal heart rhythms in individuals with a disorder called long QT syndrome. People with this condition may suddenly black out because their heart develops prolonged electric waves that prevent blood from being pumped properly.

To investigate how mutations in the $I_{Ks}$ channel produce heart rhythm abnormalities, Liin et al. genetically engineered the egg cells of African clawed frogs to have one of eight mutant forms of the human $I_{Ks}$ channel. Studying these channels revealed that the mutations reduce how well the channels work in a wide variety of ways. However, treating the cells with a particular fatty acid helped to normalize how each of the mutant channels worked. Therefore, variants of the fatty acid could potentially form a useful treatment for people with heart rhythm problems caused by mutations in the $I_{Ks}$ channel.

More studies are needed to confirm whether the fatty acid is as effective at combating the effects of the mutations in whole hearts and animals. As ion channels related to the $I_{Ks}$ channel are found in many types of cells, it is also important to investigate whether treatment with the fatty acid could cause any side effects that affect other organs.

the $I_{Ks}$ channel as $K_V7.1$+KCNE1. $K_V7.1$ has six transmembrane segments named S1-S6 (*Liin et al., 2015*) (*Figure 1a*). S1-S4 of each $K_V7.1$ subunit forms a voltage-sensing domain where S4 is the voltage sensor with three positive gating charges. S5 and S6 from all four $K_V7.1$ subunits form the pore domain with a putative gate in S6 that needs to move to open the ion-conducting pore of the channel. KCNE1 has a single-transmembrane segment (*Figure 1a*) and is proposed to be localized in the otherwise lipid-filled space between two voltage-sensing domains of neighbouring $K_V7.1$ subunits (*Nakajo and Kubo, 2015*). Upon cardiomyocyte depolarization, the voltage sensor of $K_V7.1$ moves outward in relation to the membrane. It has been proposed that this movement of the voltage sensor is transferred to the pore domain via the S4-S5 linker and induces channel opening by moving the S6 gate (*Liin et al., 2015*).

Altogether, about 300 mutations in *KCNQ1* and *KCNE1* have been identified in patients suffering from LQTS (*Hedley et al., 2009*) (http://www.fsm.it/cardmoc/). These mutations are distributed throughout the channel sequence and are therefore likely to cause channel dysfunction by different mechanisms, which are, however, largely unknown. Potential mechanisms for $K_V7.1$+KCNE1 channel loss of function by a mutation could, for example, be interference with voltage sensor movement, gate opening, or membrane expression. LQTS is today treated with drugs that prevent the triggering of arrhythmic activity, such as beta-blockers, or with arrhythmia-terminating implantable cardioverter defibrillator (*Hedley et al., 2009*). A different treatment strategy for LQTS caused by loss-of-function mutations in the $K_V7.1$+KCNE1 channel would be to pharmacologically augment the $K_V7.1$+KCNE1 channel function of these LQTS mutants, thereby shortening the prolonged QT interval and lower the risk of arrhythmia development. However, there is currently no clinically approved $K_V7.1$+KCNE1 channel activator.

In this study, we investigate the biophysical properties and potential mechanism of action of LQTS-associated $K_V7.1$+KCNE1 channel mutations and test the ability of the fatty acid analogue N-arachidonoyl taurine (N-AT) to restore the function of these mutants.

We selected eight mutations of residues mutated in patients with LQTS located in different segments of the $K_V7.1$+KCNE1 channel and that were previously shown to form active channels (*Bianchi et al., 2000*; *Yamaguchi et al., 2003*; *Eldstrom et al., 2010*; *Henrion et al., 2009*;

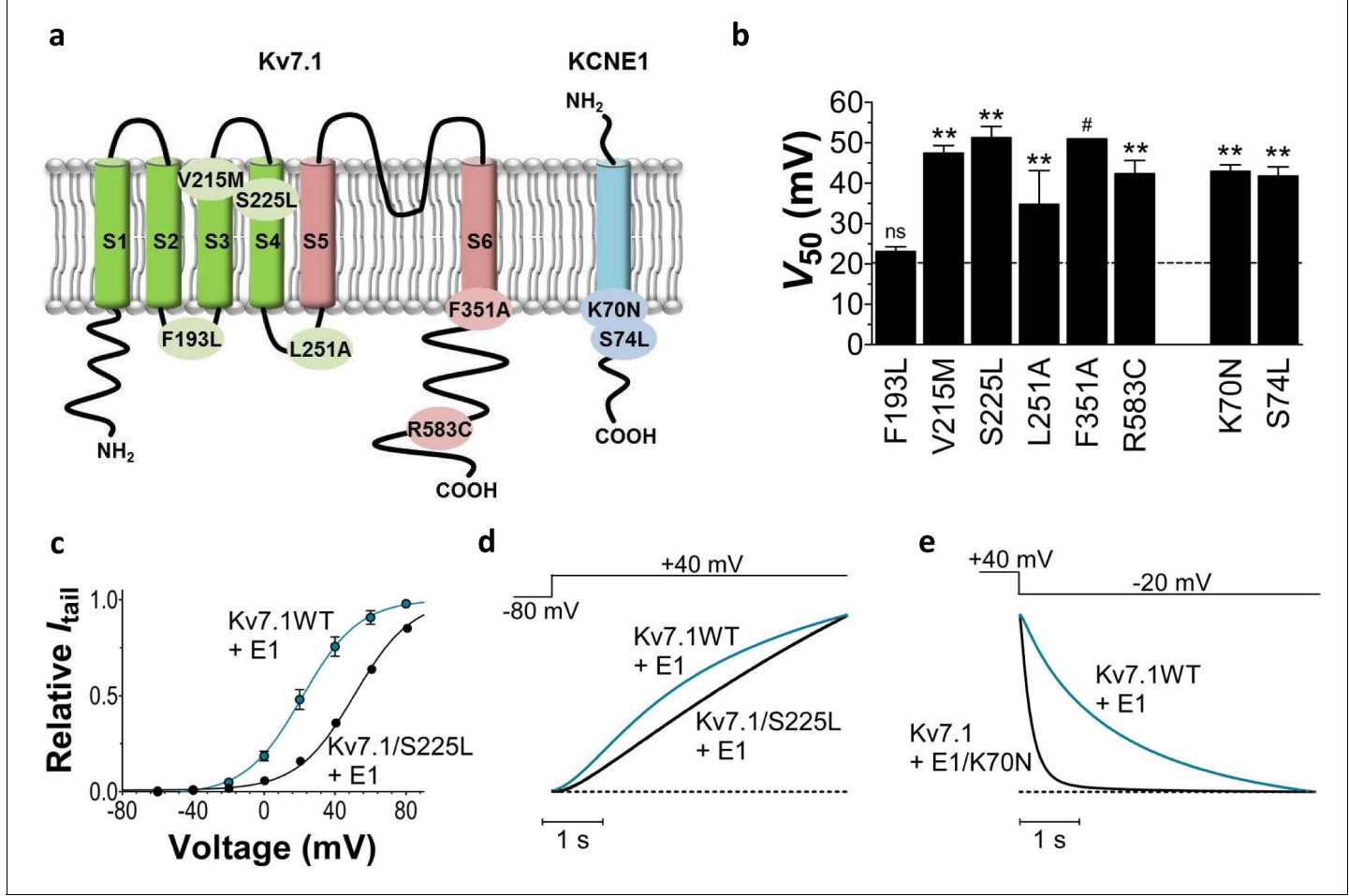

**Figure 1.** Biophysical properties of LQTS and LQTS-like $K_V7.1$+KCNE1 channel mutants expressed in *Xenopus* oocytes. (a) Topology of $K_V7.1$ and KCNE1, and position of tested LQTS and LQTS-like mutants. (b) $G(V)$ midpoints ($V_{50}$) from the Boltzmann fits for mutants co-expressed with KCNE1. $n = 5–11$. Data as mean ± SEM. The statistics represent one-way ANOVA with Dunnett's Multiple Comparison Test to compare the mutants to wild-type $K_V7.1$+KCNE1; **$p<0.01$; ns is $p\geq0.05$. # denotes lowest estimate. Dashed line denotes wild-type $V_{50}$. (c) Representative example of $K_V7.1$/S225L +KCNE1 $G(V)$ (black line and symbols) compared to wild-type $K_V7.1$+KCNE1 (blue line and symbols, mean ± SEM, $n = 5$). (d–e) Representative example of $K_V7.1$/S225L+KCNE1 opening kinetics and $K_V7.1$+KCNE1/K70N closing kinetics (black lines) compared to wild-type $K_V7.1$+KCNE1 (blue lines).

The following figure supplements are available for figure 1:

**Figure supplement 1.** $K_V7.1$/F351S mutant expressed in *Xenopus* oocytes.

**Figure supplement 2.** $V_{50}$ of LQTS and LQTS-like $K_V7.1$ mutants expressed in *Xenopus* oocytes.

**Figure supplement 3.** $K_V7.1$/R583C mutant expressed in *Xenopus* oocytes.

**Figure supplement 4.** Comparison of current amplitude of wild-type $K_V7.1$+KCNE1 and LQTS and LQTS-like mutants when expressed in *Xenopus* oocytes.

*Yang et al., 2013*; *Yang et al., 2002*; *Harmer et al., 2010*; *Splawski et al., 1997*). We measure the movement of the S4 voltage sensor in selected mutants using voltage clamp fluorometry to further our understanding of the molecular mechanisms underlying the defects caused by the diverse mutations. We find that the eight LQTS-associated mutations affect the voltage dependence and/or closing kinetics, in some cases by different molecular mechanisms. Moreover, we find that N-AT restores much of the channel activity in these eight LQTS-associated $K_V7.1$+KCNE1 mutants. This suggests

that N-AT may function as a general activator of $K_V7.1$+KCNE1 channels with diverse mutational defects.

## Results

### LQTS mutants show altered biophysical properties

We first study the biophysical properties of six point mutations in $K_V7.1$ (F193L, V215M, S225L, L251P, F351S, R583C), and two in KCNE1 (K70N, S74L) identified in patients with LQTS (*Yamaguchi et al., 2003*; *Yang et al., 2002*; *Splawski et al., 1997*; *Priori et al., 1999*; *Napolitano et al., 2005*; *Lai et al., 2005*) (*Figure 1a*). As L251P and F351S did not produce functional channels (*Napolitano et al., 2005*; *Deschenes et al., 2003*) (*Figure 1—figure supplement 1*), we engineered the milder L251A and F351A mutants instead. L251A and F351A will be referred to as 'LQTS-like mutants'. When expressed alone in *Xenopus* oocytes, all investigated $K_V7.1$ mutants, except F193L and V215M, display a shifted conductance *versus* voltage curve ($G(V)$) compared to the wild-type $K_V7.1$ channel (*Figure 1—figure supplement 2*; *Supplementary file 1*). S225L, L251A and F351A shift the $G(V)$ towards positive voltages compared to wild-type $K_V7.1$. In contrast, R583C shifts the half-maximal activation, $V_{50}$, ~10 mV towards negative voltages compared to wild-type $K_V7.1$. This apparent negative shift is likely caused by the pronounced inactivation of the R583C mutant (*Figure 1—figure supplement 3a*), which is seen to a considerable smaller extent in the other $K_V7.1$ mutants and wild-type $K_V7.1$ (inset in *Figure 1—figure supplement 3a*). When a fraction of the channels are released from inactivation, by introducing a brief hyperpolarizing pulse between the test pulse and the tail pulse, R583C has a $V_{50}$ fairly comparable to wild-type $K_V7.1$ (*Figure 1—figure supplement 3b*).

When the $K_V7.1$ mutants are co-expressed with KCNE1, all $K_V7.1$ and KCNE1 mutants except $K_V7.1$/F193L+KCNE1 have a $G(V)$ that is shifted towards positive voltages compared to the wild-type $K_V7.1$+KCNE1 channel (*Figure 1b*). $K_V7.1$/F351A causes the most dramatic change by shifting $V_{50}$ more than +30 mV. We are therefore only able to record the foot of the $G(V)$ curve of $K_V7.1$/F351A+KCNE1, and a shift in $V_{50}$ of +30 mV is a lower estimate of the change in $V_{50}$ ($\Delta V_{50}$). One of the other mutants with dramatically shifted $G(V)$ is $K_V7.1$/S225L+KCNE1. $V_{50}$ for $K_V7.1$/S225L +KCNE1 is shifted almost +30 mV compared to wild-type $K_V7.1$+KCNE1 (*Figure 1c*; *Supplementary file 1*). S225L also slows down $K_V7.1$+KCNE1 channel opening kinetics (p<0.01; *Figure 1d*; *Supplementary file 1*). All mutations, except for L251A, accelerate channel closing kinetics compared to wild-type $K_V7.1$+KCNE1 (*Supplementary file 1*). K70N has the most dramatic effect on $K_V7.1$+KCNE1 channel closing by accelerating the closing kinetics by approximately a factor of 5 (*Figure 1e*; *Supplementary file 1*). When comparing the amplitude of $K^+$ currents generated by these mutants with the current amplitude of the wild-type $K_V7.1$+KCNE1 channel in the same batch of oocytes, we note that all mutants generate smaller currents than wild-type over a large voltage range (*Figure 1—figure supplement 4*). Although defective trafficking may contribute to these reduced currents in *Xenopus* oocytes, the current amplitudes for most mutants matches fairly well with the predicted current amplitude from channels with $G(V)$ curves shifted towards positive voltages as observed for these mutants (*Figure 1—figure supplement 4a*), suggesting that the reduced current amplitudes in *Xenopus* oocytes are mainly a result of gating defects (and not trafficking defects).

To summarize, all mutations change channel function by altering the voltage dependence of opening and/or the kinetics of opening and/or closing. Reduced function of the $K_V7.1$+KCNE1 channel induced by these LQTS and LQTS-like mutations may largely be explained by the right-shifted $G(V)$ and the faster closing kinetics caused by these mutations. F193L does not alter the $G(V)$, but speeds up $K_V7.1$+KCNE1 channel closing by a factor of 2 (*Supplementary file 1*). These results are consistent with previous reported findings for some of these mutants (*Bianchi et al., 2000*; *Yamaguchi et al., 2003*; *Eldstrom et al., 2010*; *Henrion et al., 2009*; *Yang et al., 2013*; *Yang et al., 2002*; *Harmer et al., 2010*).

### Heterozygous expression reduces LQTS mutant severity

Patients with LQTS mutations can be either homozygous or heterozygous for the mutation. To mimic heterozygous expression, we co-inject the mutated $K_V7.1$ subunit and KCNE1 subunit together with

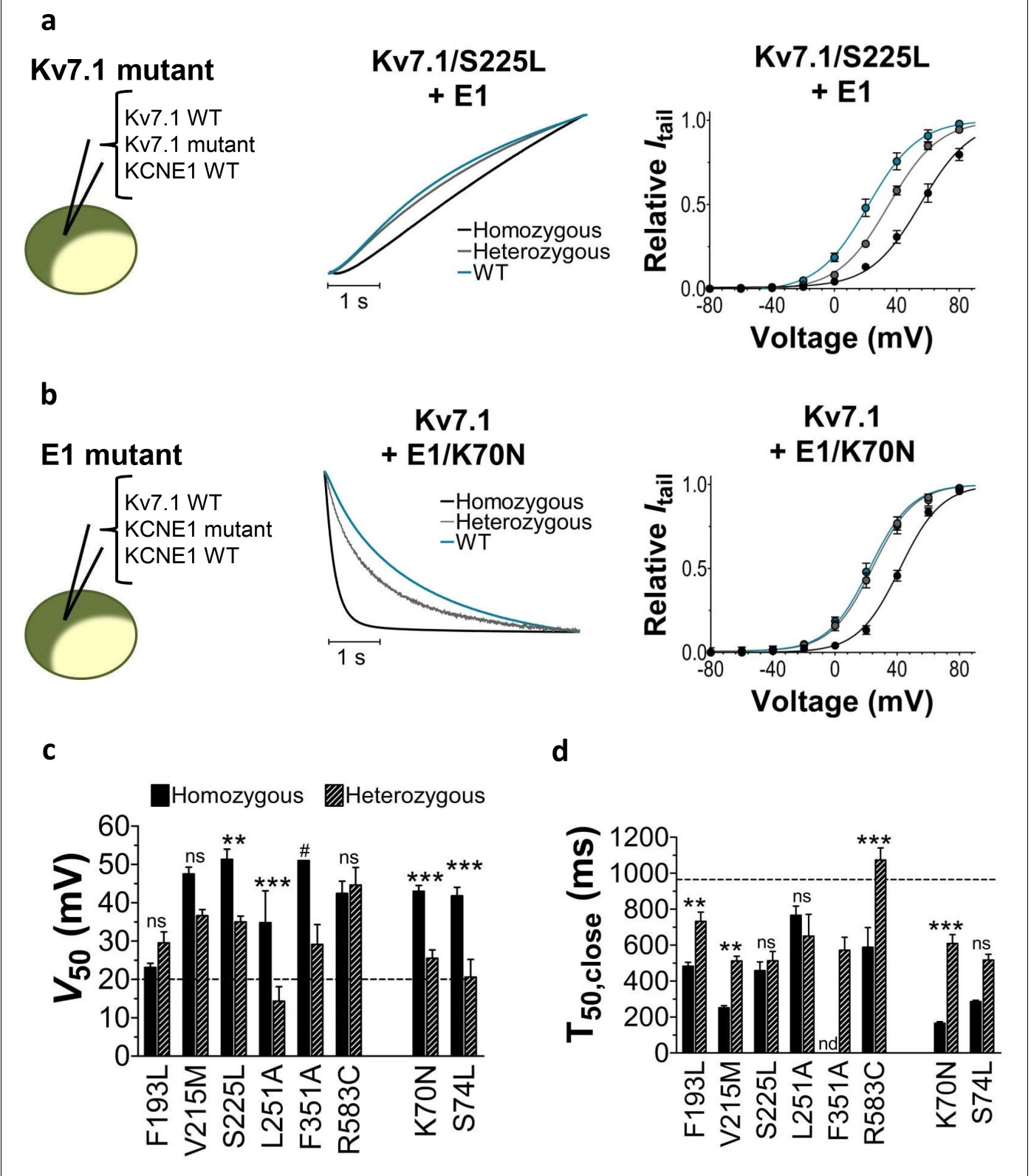

**Figure 2.** Comparison of homozygous and heterozygous expression of LQTS and LQTS-like mutants. (a–b) Representative example of kinetics (middle panel) and $G(V)$ (right panel) for homozygous expression and heterozygous expression of S225L (a) and K70N (b). Currents in response to steps from –80 mV to +40 mV (a, middle pane) and from +40 mV to –20 mV (b, middle panel). Homozygous expression (black), heterozygous expression (gray), and $K_V7.1$+KCNE1 wild-type (blue). $n$ = 7–13. (c–d) Summary of $V_{50}$ (c) and $T_{50}$ for closing (d) for homozygous and heterozygous expression. Data as mean ±

*Figure 2 continued on next page*

Figure 2 continued

SEM. *n* = 5–13. The statistics represent one-way ANOVA with pair-wise Bonferroni's Test to compare homozygous and heterozygous expression; **p<0.01; ***p<0.001; ns is p≥0.05. # denotes lowest estimate. Not determined (nd). The statistics was not calculated for F351A. Dashed lines denote corresponding values for wild-type $K_V7.1$+KCNE1.

the wild-type $K_V7.1$ subunit (or wild-type KCNE1 subunit for KCNE1 mutants) (cartoon in *Figure 2*). We refer to this as heterozygous expression. *Figure 2a–b* compares the homozygous expression ($K_V7.1^{wt}$+KCNE1$^{mut}$ or $K_V7.1^{mut}$+KCNE1$^{wt}$) with heterozygous expression ($K_V7.1^{wt}$+$K_V7.1^{mut}$+KCNE1$^{wt}$ or $K_V7.1^{wt}$+KCNE1$^{wt}$+KCNE1$^{mut}$) for $K_V7.1$/S225L (*Figure 2a*) and KCNE1/K70N (*Figure 2b*). Both of these examples show that heterozygous expression generates channels with more wild-type like opening or closing kinetics and *G*(*V*) compared to homozygous expression of the mutant subunit. A milder biophysical phenotype upon heterozygous expression is generally seen for the LQTS and LQTS-like mutants in terms of *G*(*V*), current amplitude, and/or closing kinetics (*Figure 2c–d*, *Figure 1—figure supplement 4*, *Supplementary file 2*). This milder phenotype indicates that the wild-type subunit can partly restore $K_V7.1$+KCNE1 function. Alternatively, for mutants with a *G*(*V*) that is very shifted to positive voltages (e.g. F351A), it may be that channel complexes that contain the mutated subunits are largely out of the physiological voltage range and therefore do not contribute substantially to the recorded current. Also, for mutants with low membrane expression (e.g. possibly F193L [*Yamaguchi et al., 2003*]), it may be that channels containing the wild-type subunit are favoured so that in most $K_V7.1$+KCNE1 channel complexes the majority (or all) of the subunits will be wild-type subunits.

## Different mutants display different fluorescence versus voltage profiles

Although most of the mutations shift channel voltage dependence and affect channel closing kinetics, the underlying mechanism of mutation-induced changes in $K_V7.1$+KCNE1 channel function is most likely different for different mutations. For instance, mutations located in S5 and S6 (e.g. F351A) may mainly affect gate movement, while mutations in S1–S4 (e.g. S225L) are more likely to affect voltage sensor movement. To explore whether different mutations interfere with different gating transitions, we use voltage clamp fluorometry, in which the movement of the voltage sensor in $K_V7.1$ can be tracked by the fluorescence change from the fluorescent probe Alexa-488-maleimide attached to G219C in the S3-S4 loop (referred to as G219C*) (*Barro-Soria et al., 2014*; *Osteen et al., 2010*; *Osteen et al., 2012*). Voltage sensor movement (measured by fluorescence) and gate movement (measured by ionic currents) are then monitored under two-electrode voltage clamp. The $K_V7.1$/G219C* construct by itself or co-expressed with KCNE1 gives voltage-dependent fluorescence changes (*Figure 3a*). As previously reported, the fluorescence *versus* voltage (*F*(*V*)) curve of $K_V7.1$/G219C* correlates well with the *G*(*V*) curve (*Figure 3a*, left panel), while the *F*(*V*) curve of $K_V7.1$/G219C*+KCNE1 is divided into two components (*Figure 3a*, right panel) (*Barro-Soria et al., 2014*; *Osteen et al., 2010*; *Osteen et al., 2012*). For $K_V7.1$/G219C*+KCNE1, the first fluorescence component (*F1*) has been suggested to represent the main voltage sensor movement and the second fluorescence component (*F2*) to be correlated with gate opening (*Barro-Soria et al., 2014*). We introduce G219C into $K_V7.1$/S225L and $K_V7.1$/F351A. The *G*(*V*) curves of both $K_V7.1$/G219C*/S225L and $K_V7.1$/G219C*/F351A are shifted towards more positive voltages compared to the wild-type channel, but the *F*(*V*) curves are differentially affected by the two mutations (*Figure 3b–c*, left panels). For $K_V7.1$/G219C*/S225L, the *F*(*V*) curve is shifted to a similar extent as the *G*(*V*) curve, while for $K_V7.1$/G219C*/F351A, the *F*(*V*) curve is shifted to a considerably smaller extent (*Osteen et al., 2010*). When these mutants are co-expressed with KCNE1, we observe different effects on the voltage dependence of the two fluorescent components *F1* and *F2* induced by the mutations. The S225L mutation primarily shifts *F1* towards positive voltages so that *F1* and *F2* of $K_V7.1$/G219C*/S225L+KCNE1 are hardly distinguishable in the *F*(*V*) curve (*Figure 3b*, right panel). In contrast, the F351A mutation primarily shifts *F2* towards positive voltages so that *F1* and *F2* are clearly separated (*Figure 3c*, right panel). Thus, S225L and F351A seem to shift the *G*(*V*) curve of $K_V7.1$+KCNE1 towards positive voltages by interfering with different gating transitions.

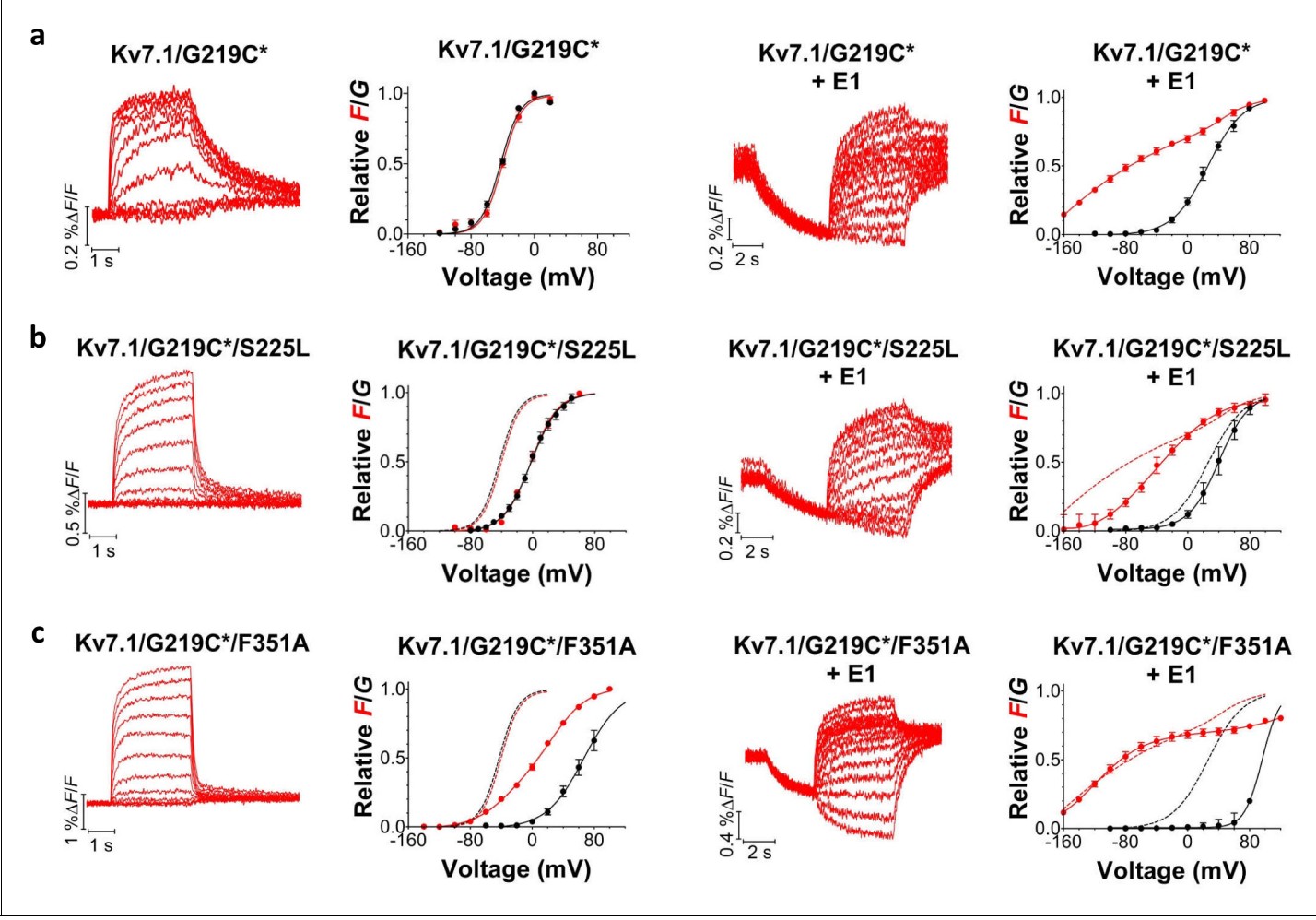

**Figure 3.** Voltage-clamp fluorometry recordings of wild-type and mutated $K_V7.1$+KCNE1 channels. (a-c) Representative fluorescence traces and mean $F$ (V)/G(V) curves for $K_V7.1$/G219C* (a), S225L (b), and F351A (c). Left panels without KCNE1 and right panels with KCNE1. The holding voltage is –80 mV, the pre-pulse –120 mV for 2 s (left panels) and –160 mV for 5 s (right panels), and test voltages between –140 and +80 mV for 3 s (left panels) and between –160 and +80 mV for 5 s (right panels) in 20 mV increments. The tail voltage is –80 mV (left panels) and −40 mV (right panels). For $K_V7.1$/ G219C*/F351A+KCNE1, the pre-pulse is –120 mV for 3 s, and test voltages ranging between –160 and +100 mV. The bottom of the fit of the $K_V7.1$/ G219C*/S225L+KCNE1 $F$(V) curve (which saturates fairly well at negative voltages) is set to 0 in the normalized $F$(V) curves in the right panels. The $F1$ amplitude of $K_V7.1$/G219C*/F351A+KCNE1 is normalized to the $F1$ amplitude of wild-type. Data as mean ± SEM. $n$ = 4–14. The dashed lines in (b) and (c) denote $F$(V) (red) and $G$(V) (black) for wild-type (from a).
The following figure supplements are available for figure 3:

**Figure supplement 1.** Kinetic models for $K_V7.1$ and $K_V7.1$+KCNE1 channel gating.

**Figure supplement 2.** Simulations of wild-type and mutant $K_V7.1$ and $K_V7.1$+KCNE1 channels reproduce currents and fluorescence.

**Figure supplement 3.** Voltage-clamp fluorometry recordings of the $K_V7.1$/G219C*/F351L mutant with and without KCNE1 co-expressed.

## Kinetic modeling recapitulates experimental findings

To further explore the different effects of S225L and F351A in the voltage-clamp fluorometry experiments, we use two kinetic models previously developed to reproduce the currents and fluorescence from $K_V7.1$/G219C* (*Osteen et al., 2012*) and $K_V7.1$/G219C*+KCNE1 channels (*Barro-Soria et al., 2014*), respectively. The $K_V7.1$/G219C* model is an allosteric model with 10 states (*Figure 3—figure supplement 1a*), where the horizontal transition is the main S4 movement (which generates the main

fluorescence component $F1$) and the vertical transition is channel opening accompanied by an additional smaller S4 movement (that generates a smaller additional fluorescence component $F2$) (Osteen et al., 2012; Zaydman et al., 2014). The $K_V7.1/G219C^*$ model allows for channel opening after only a subset of four S4s are activated, which thereby generates $F(V)$ and $G(V)$ that are close in the voltage dependence (reference (Osteen et al., 2012); and Figure 3—figure supplement 2a). The $K_V7.1/G219C^*$+KCNE1 model has 6 states (Figure 3—figure supplement 1b), where the horizontal transition is the main S4 movement (which generates the main fluorescence component $F1$) and the vertical transition is channel opening accompanied by an additional smaller S4 movement (that generates a smaller additional fluorescence component $F2$) (Osteen et al., 2012; Zaydman et al., 2014). The $K_V7.1/G219C^*$+KCNE1 model only allows for channel opening after all four S4s are activated, which thereby generates $F(V)$ and $G(V)$ that are separated in voltage dependence (reference [Barro-Soria et al., 2014]; and Figure 3—figure supplement 2a).

Using these models, we can reproduce the main features of the fluorescence and currents from $K_V7.1/G219C^*/S225L$ and $K_V7.1/G219C^*/S225L$+KCNE1 by only shifting the main voltage sensor movement by +50 mV in both models (Figure 3—figure supplement 2b), as if the S225L mutation mainly affects the main S4 movement. In the $K_V7.1$ model, shifting the main voltage sensor movement by +50 mV shifts both the $G(V)$ and $F(V)$ curves by +35–40 mV, similar to the effect induced by the S225L mutation in the experimental data. In the $K_V7.1$+KCNE1 model, shifting the main voltage sensor movement by +50 mV results in that the $F1$ and $F2$ components overlap in voltage, such that it is hard to distinguish the two components, and that the $G(V)$ is shifted by +10 mV. Both effects are similar to the effects induced by the S225L mutation in the experimental data (cf. Figure 3b).

We can reproduce the main features of the fluorescence and currents from $K_V7.1/G219C^*/F351A$ and $K_V7.1/G219C^*/F351A$+KCNE1 by only shifting the voltage dependence of the opening transition by +140 mV in both models (Figure 3—figure supplement 2c), as if the F351A mutation mainly affects the opening transition. In the $K_V7.1$ model, shifting the opening transition by +140 mV shifts the $G(V)$ by +100 mV whereas the $F(V)$ is shifted less and has a shallower slope, similar to the effects induced by the F351A mutation in the experimental data. In the $K_V7.1$+KCNE1 model, shifting the opening transition by +140 mV results in that the $F1$ and $F2$ components are further separated in voltage and that the $G(V)$ is shifted by +100 mV. Both effects are similar to the effects induced by the F351A mutation in the experimental data (cf. Figure 3c).

In summary, our voltage-clamp fluorometry experiments together with kinetic modeling are compatible with a model in which the S225L mutation primarily interferes with the main S4 movement, whereas the F351A mutation interferes with later gating transitions associated with pore opening. One note of caution is that the interpretation of the mutational effects is dependent on the models used for the wild-type channels. Other models for $K_V7.1$ and $K_V7.1$+KCNE1 channels have been proposed (Zaydman et al., 2014; Ruscic et al., 2013), but these have not been as extensively tested or developed as our models. Although other alternative mechanisms for the effects of these mutations are possible, the different impacts of S225L and F351A on the fluorescence versus voltage relationships suggest that these mutations introduce distinct molecular defects.

## N-AT enhances the activity of all tested LQTS and LQTS-like mutants

We previously observed that the effect of regular polyunsaturated fatty acids, such as docosahexaenoic acid, on $K_V7.1$ is impaired by co-expression with the KCNE1 subunit (Liin et al., 2015). In contrast, we found that the PUFA analogue N-arachidonoyl taurine (N-AT, structure in Figure 4) retained its ability to activate the $K_V7.1$ channel also in the presence of KCNE1. N-AT activated the wild-type $K_V7.1$+KCNE1 by shifting the $G(V)$ roughly –30 mV (Liin et al., 2015) (Figure 4—figure supplement 1). The magnitude of this N-AT-induced shift is comparable to, but in the opposite direction, to the $G(V)$ shifts observed for several of the LQTS and LQTS-like mutants. We therefore here test the ability of N-AT to enhance the function of the eight $K_V7.1$+KCNE1 mutant channels. Figure 4a–b shows representative effects of 7–70 μM N-AT on $K_V7.1/S225L$+KCNE1. 70 μM N-AT increases current amplitude by a factor of 16 at +20 mV (Figure 4a) and shifts the $G(V)$ curve by about –50 mV (Figure 4b, Supplementary file 3). Steady state of N-AT effects is reached within a few minutes (Figure 4—figure supplement 2). We note a small instantaneous 'leak' component in the 70 μM N-AT trace of $K_V7.1/S225L$+KCNE1 (Figure 4a). This leak component in $K_V7.1/S225L$+KCNE1 is observed also in the absence of N-AT, but at more positive voltages (Figure 4—figure supplement 3). We do not observe this leak component in wild-type $K_V7.1$+KCNE1 upon application

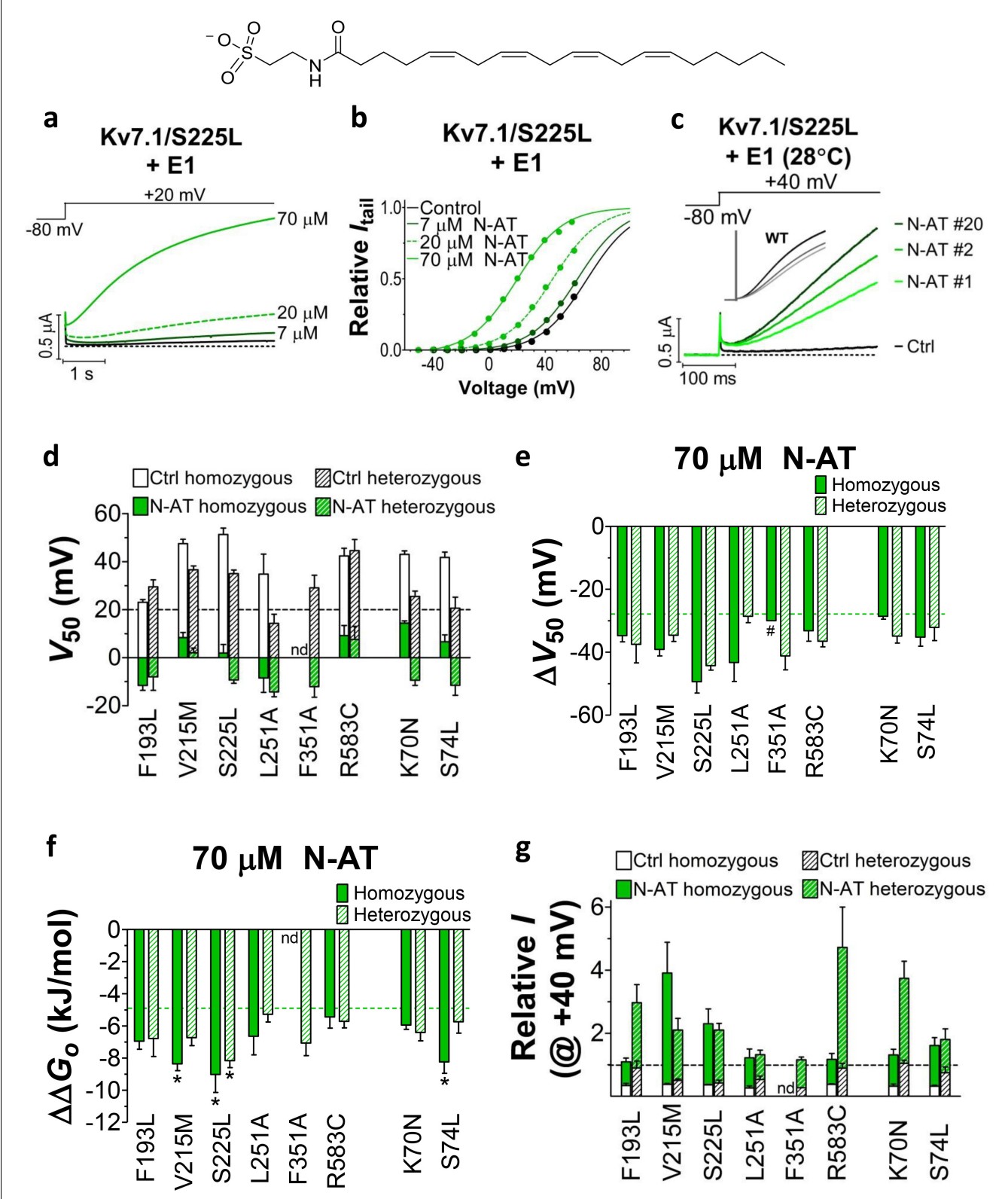

**Figure 4.** Effect of N-AT on LQTS and LQTS-like mutants. All these experiments are done in the presence of KCNE1. Structure of N-AT is shown. (a–b) Representative effect of 7–70 μM N-AT on current amplitude (a) and $G(V)$ (b) of $K_V7.1/S225L+KCNE1$. Dashed line in (a) denotes 0 μA. (c) Representative currents generated by $K_V7.1/S225L+KCNE1$ during pulsing at 1 Hz and +28°C in control solution (black) and after the cell had been bathed continuously in 70 μM N-AT (light to dark green, # denotes sweep order). Inset: corresponding currents from wild-type $K_V7.1+KCNE1$ scaled similarly as $K_V7.1/S225L$

*Figure 4 continued on next page*

*Figure 4 continued*

+KCNE1. Light grey trace denotes sweep #1, grey trace denotes sweep #2, and dark grey trace denotes sweep #20. (d) Summary of $V_{50}$ for LQTS and LQTS-like mutants before and after 70 μM N-AT application. Dashed line denotes $V_{50}$ for wild-type $K_V7.1$+KCNE1. (e–f) Summary of $\Delta V_{50}$ (e) and $\Delta\Delta G_o$ (f) for LQTS and LQTS-like mutants induced by 70 μM N-AT. # denotes an approximation. Dashed lines denote corresponding $\Delta V_{50}$ and $\Delta\Delta G_o$ induced by 70 μM N-AT for wild-type $K_V7.1$+KCNE1. The statistics in (f) represent one-way ANOVA with Dunnett's Multiple Comparison Test to compare the N-AT-induced change in $\Delta\Delta G_o$ of mutants to N-AT-induced change in $\Delta\Delta G_o$ of wild-type $K_V7.1$+KCNE1; *p≤0.05. Only significant differences shown in (f), other comparisons have p>0.05. (g) Estimate of the ability of 70 μM N-AT to restore LQTS and LQTS-like mutant current amplitude at +40 mV. The mean N-AT induced increase in current amplitude for each mutant (from *Figure 4—figure supplement 4b*) is multiplied with the control amplitude for each mutant (from *Figure 1—figure supplement 4d*). Not determined (nd). Data as mean ± SEM. *n* = 5–12. Dashed line denotes relative wild-type $K_V7.1$+KCNE1 current amplitude in control solution (i.e. without N-AT).

The following figure supplements are available for figure 4:

**Figure supplement 1.** N-AT effect on wild-type $K_V7.1$+KCNE1 expressed in *Xenopus* oocytes.

**Figure supplement 2.** The time course of N-AT wash-in on $K_V7.1$/S225L+KCNE1 expressed in *Xenopus* oocytes.

**Figure supplement 3.** 'Leak' component of $K_V7.1$/S225L+KCNE1.

**Figure supplement 4.** Effect of N-AT on current amplitude of LQTS and LQTS-like mutants.

of N-AT (*Figure 4—figure supplement 1a*), which suggests that this phenomenon is associated with the S225L mutation. The human ventricular action potential has a duration of about 300–400 ms and a systolic voltage range of about 0 to +40 mV (*O'Hara et al., 2011*; *Piacentino et al., 2003*). To test the behaviour of the S225L mutation during shorter stimulating pulses, we apply repetitive 300 ms pulses to +40 mV at a frequency of 1 Hz and at 28°C (37°C was not tolerated by the oocytes). In response to this protocol, the $K_V7.1$/S225L+KCNE1 channel barely opens and thus generates only minor currents (*Figure 4c*). In contrast, we observe large $K_V7.1$/S225L+KCNE1 currents upon application of 70 μM N-AT (*Figure 4c*). N-AT also restores the gradual increase in current amplitude during repetitive pulsing seen experimentally (inset in *Figure 4c*) and in computer simulations (*Silva and Rudy, 2005*) for the wild-type $K_V7.1$+KCNE1 channel.

Further testing of N-AT show that 70 μM N-AT shifts the $G(V)$ curve of all tested mutants by 30–50 mV towards more negative voltages (*Figure 4d–e*, *Supplementary file 3*). The $G(V)$ curve of wild-type $K_V7.1$+KCNE1 is shifted by –27.0 ± 2.5 mV (*Liin et al., 2015*). Thus, 70 μM N-AT completely corrects the positive $G(V)$ shifts induced by the mutations so that in the presence of N-AT the $G(V)$ is similar to or shifted negative compared to the $G(V)$ of the wild-type $K_V7.1$+KCNE1 channel (*Figure 4d*, F351A homozygous expression was not included in this analysis because of the very shifted $G(V)$ curve of this mutant). The $G(V)$ of mutants is shifted about equally by N-AT for homozygous and heterozygous expression (*Figure 4e*). The slope of the $G(V)$ curve varies slightly (10.4 to 16.3) among the mutants (*Supplementary file 3*). To correct for this difference in slope and to better compare the functional effect of N-AT-induced $G(V)$ shifts on the different mutants, we also calculate the change in Gibbs free energy for channel opening ($\Delta\Delta Go$) that 70 μM N-AT induces. 70 μM N-AT reduces the energy required to open the channel by 5.3–9.0 kJ/mol depending on mutant (4.9 ± 0.7 kJ/mol (n = 5) for wild-type) (*Figure 4f*). To estimate the functional effect of N-AT on the $K_V7.1$+KCNE1 current amplitude of each mutant, we calculate the ratio of the current amplitude at the end of the 5 s test pulse before and after application of N-AT at +20 and +40 mV. The 5 s voltage pulse to +20 mV (or + 40 mV) at room temperature was chosen to make the $K_V7.1$+KCNE1 channel activate to a similar extent as during a ventricular action potential (300–400 ms) at body temperature (note that $K_V7.1$+KCNE1 channels have a relatively high $Q_{10}$ of around 5–7.5 [*Busch and Lang, 1993*; *Seebohm et al., 2001*]). 70 μM N-AT increases the current amplitude of all mutants at these voltages (*Figure 4—figure supplement 4a–b*, *Supplementary file 3*). As expected, current amplitude is most increased for those mutants that have the most shifted $G(V)$ curve towards more positive voltages (e.g. V215M and S225L). This is because these mutants are still at the foot of their $G(V)$ curve at +20 and +40 mV and a N-AT-induced shift towards more negative voltages results in a relatively larger increase in the current amplitude. By multiplying these relative N-AT-

induced increases in current amplitude with the relative current amplitude of each mutant (compared to wild-type $K_V7.1$+KCNE1 channels, from *Figure 1—figure supplement 4c–d*), we observe that 70 µM N-AT compensates fairly well (or overcompensates) for the mutation-induced reduction in current amplitude (*Figure 4g*, *Figure 4—figure supplement 4c*). Moreover, for all mutant and wild-type $K_V7.1$+KCNE1 channels, 70 µM N-AT speeds up the opening kinetics at +40 mV by a factor of 1.3–2.5 (*Supplementary file 3*). 70 µM N-AT also slows down the closing kinetics for most mutants and wild-type $K_V7.1$+KCNE1 (*Supplementary file 3*). For F351A heterozygous expression and R583C homozygous expression, 70 µM N-AT restores the closing kinetics so that the closing kinetics is not statistically different (p>0.05) from wild-type $K_V7.1$+KCNE1 closing kinetics (737 ± 62 ms and 833 ± 74 ms, respectively compared to 967 ± 47 ms for wild-type). In the presence of KCNE1, channels made with F193L heterozygous expression, L251A homozygous expression, and R583C heterozygous expression have wild-type like closing kinetics already before application of N-AT.

## N-AT affects both S4 movement and gate opening in mutants

We next use voltage clamp fluorometry on $K_V7.1$/G219C*/S225L+KCNE1 and $K_V7.1$/G219C*/F351A +KCNE1 to explore the mechanism by which N-AT enhances the activity of two mechanistically different mutants. Surprisingly, N-AT caused a dramatic decrease in the fluorescence from Alexa488-labeled $K_V7.1$/G219C*+KCNE1 channels (*Figure 5—figure supplement 1a*). In contrast, N-AT did not decrease the fluorescence from Alexa488-labeled $K_V7.1$/G219C* channels nor did high concentrations of taurine decrease the fluorescence from unbound Alexa488 (even up to concentrations of 0.5 M taurine; *Figure 5—figure supplement 1b*), suggesting that N-AT is not a collisional quencher of Alexa488. The mechanism of the N-AT-induced decrease of fluorescence from Alexa488-labeled $K_V7.1$/G219C*+KCNE1 channels is not clear, but could be due to N-AT inducing a conformational change in KCNE1 or $K_V7.1$ that brings a quenching residue close to Alexa488.

Due to the dramatic decrease in the fluorescence signal from Alexa488-labeled $K_V7.1$/G219C* +KCNE1 channels, we have to normalize the $F(V)$ curves obtained in N-AT to the amplitude of the $F(V)$ in control solutions. With this normalization, voltage clamp fluorometry experiments on $K_V7.1$/G219C*/S225L+KCNE1 indicate that N-AT shifts both the voltage dependence of the first part (which represents $F1$) and the second part (which represents $F2$) of the $F(V)$ curve towards more negative voltages (*Figure 5—figure supplement 1c*). However, due to the not completely saturating $F(V)$ for $K_V7.1$/G219C*/F351A+KCNE1, we are unable to reliably normalize the $F(V)$ curves in the presence of N-AT to the control $F(V)$ curves. We instead explore the effect of N-AT on the kinetics of the two fluorescence components: $F1$, which is seen as a fast fluorescence change at negative voltages, and $F2$, which is seen as a slow fluorescence change on top of the $F1$ component at positive voltages (*Barro-Soria et al., 2014*). $F1$ correlates with the measured gating currents in $K_V7.1$+KCNE1 channels (and the initial delay in the $K_V7.1$+KCNE1 ionic currents), whereas $F2$ correlates with the opening of $K_V7.1$+KCNE1 channels (*Barro-Soria et al., 2014*). For both mutants, 70 µM N-AT speeds up $F1$ kinetics (*Figure 5a,d*, measured at –40 mV where virtually no channels open and the fluorescence is mainly composed of $F1$). Numeric values for N-AT effects on channel kinetics are summarized in *Figure 5f*. Moreover, N-AT accelerates the channel opening kinetics (*Figure 5b,e*) and both the $F1$ and $F2$ fluorescence components at +80 mV for $K_V7.1$/G219C*/S225L+KCNE1 (*Figure 5f*). The change in the $F2$ component is probably larger than what the fits of a double-exponential function suggest, because the slow part of the fluorescence, mainly $F2$, overlay nicely on the currents in both the presence and absence of 70 µM N-AT (*Figure 5c*, upper panel). As a control, we show that the fluorescence in N-AT does not, however, overlay the currents in control solutions and vice versa (*Figure 5c*, middle and lower panel). For $K_V7.1$/G219C*/F351A+KCNE1, the $G(V)$ curve and the $F2$ component are so shifted towards depolarizing voltages that we cannot reliably quantify the $F2$ component in our fluorescence traces. 70 µM N-AT does, however, speed up $K_V7.1$/G219C*/F351A +KCNE1 current kinetics (*Figure 5e*), which suggests that N-AT also speeds up $F2$ in $K_V7.1$/G219C*/F351A+KCNE1. Altogether, these results suggest that N-AT accelerates both conformational changes during the main gating charge movement and channel opening.

## Discussion

We show that all studied LQTS and LQTS-like mutations i) shift the $G(V)$ of $K_V7.1$+KCNE1 towards more positive voltages, and/or ii) accelerate $K_V7.1$+KCNE1 closing. This suggests that at least part

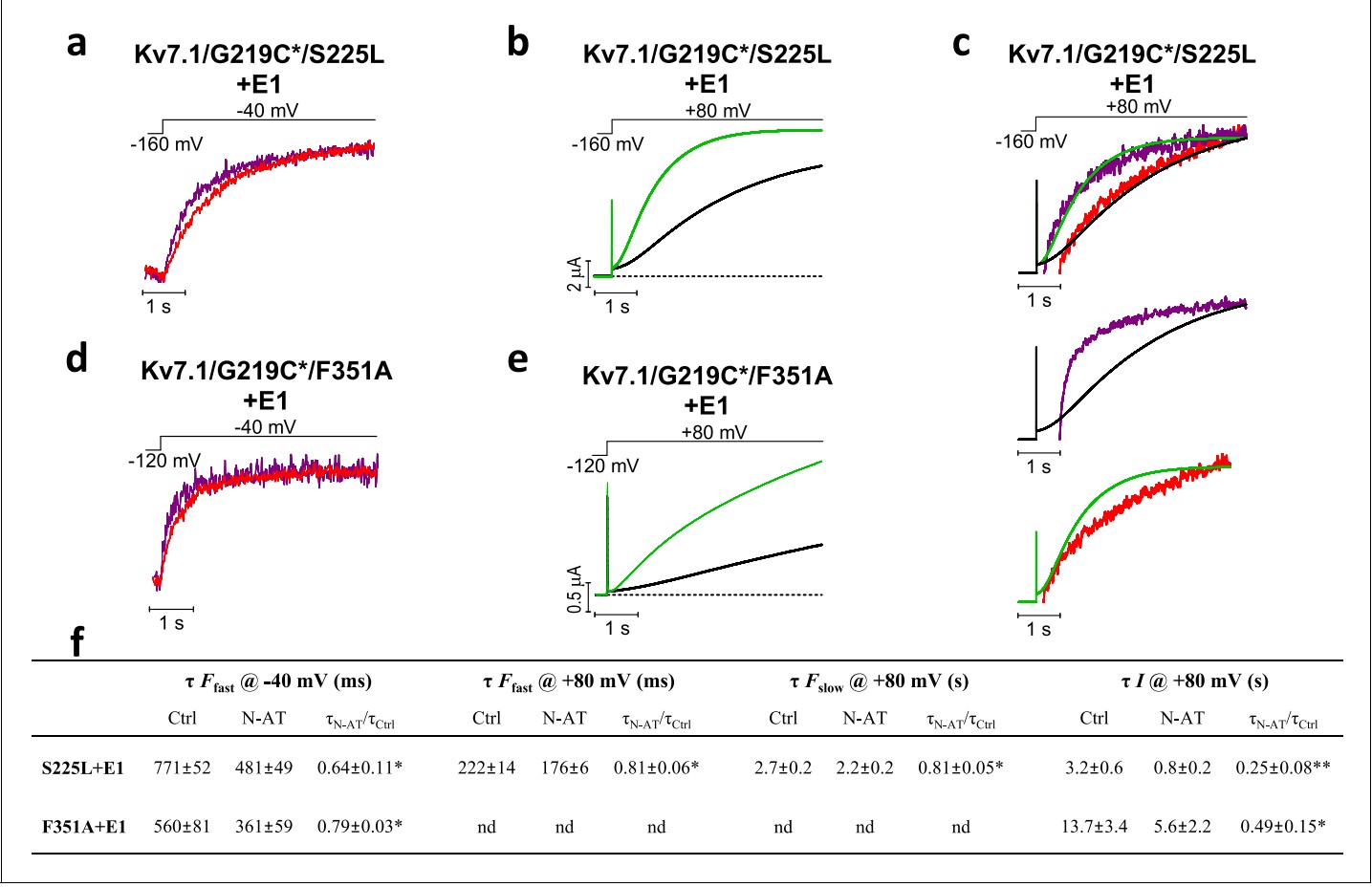

**Figure 5.** Effect of 70 μM N-AT on S4 movement and gate opening in S225L and F351A mutants. (**a–c**) Representative example of the effect of 70 μM N-AT on $F1$ kinetics (**a**), current opening kinetics (**b**), and $F2$ kinetics (**c**) in $K_V7.1/G219C*/S225L+KCNE1$. Control fluorescence (red) and current (black). N-AT fluorescence (magenta) and current (green). Top in (**c**) shows an overlay of the later part of the fluorescence (after most of $F1$ has occurred) and the later part of the currents (after the initial delay) before and after application of N-AT. Middle and lower (**c**) show that there is not a great overlap of the fluorescence in the presence of N-AT and the current in control solution (middle) or the fluorescence in control solution and the current in the presence of N-AT (lower). (**d–e**) Representative example of effect of 70 μM N-AT on $F1$ kinetics (**d**) and current opening kinetics (**e**) in $K_V7.1/G219C*/F351A+KCNE1$. Same colouring as in (**a–b**). Dashed line in (**b**) and (**e**) denotes 0 μA. Fluorescence traces and all traces in (**c**) have been normalized to better allow temporal comparison. (**f**) Summary of the effect of 70 μM N-AT on the kinetic parameters of $K_V7.1/G219C*/S225L+KCNE1$ and $K_V7.1/G219C*/F351A+KCNE1$. Kinetics of the fast ($F1$) and slow ($F2$) fluorescence components were deduced from a double-exponential function fitted to the fluorescence traces. The kinetics of currents were deduced from a single-exponential function fitted to current traces. Ratios of time constants ($\tau_{N-AT}/\tau_{Ctrl}$) were calculated pair-wise (control compared to N-AT) in each oocyte and analysed using two-tailed one sample *t*-test where ratios were compared with a hypothetical value of 1. Data as mean ± SEM. $n = 4$ (3for fluorescence kinetics for $K_V7.1/G219C*/F351A+KCNE1$). *p<0.05; **p<0.01. nd = not determined.

The following figure supplement is available for figure 5:

**Figure supplement 1.** Effect of N-AT on the $F(V)$ of $K_V7.1/G219C*/S225L$ mutant co-expressed with KCNE1 in *Xenopus* oocytes.

of the mechanism underlying the reduced ability of these mutants to generate $K^+$ currents is by altering these biophysical properties of the $K_V7.1+KCNE1$ channel. Using voltage clamp fluorometry in combination with kinetic modeling, we further suggest that these altered biophysical properties in mutants may be caused by interference with different gating transitions. Our experimental data and kinetic modeling are consistent with a model in which $K_V7.1/S225L$ primarily causes the reduced channel function by altering the main voltage sensor movement, while $K_V7.1/F351A$ alters later gating transitions associated with pore opening. The different effects of S225L and F351A on the fluorescence *versus* voltage relationships in $K_V7.1/G219C*$ and $K_V7.1/G219C*+KCNE1$ suggest that

these mutations cause channel dysfunction via different molecular mechanisms. Note that we used the LQTS-like F351A mutant, because the LQTS mutant F351S did not generate any currents (*Figure 1—figure supplement 1*). However, during the review process of this manuscript a new LQTS mutation, F351L, was found (*Vyas et al., 2016*). The current and fluorescence of this LQTS mutant is very similar to the current and fluorescence of F351A (*Figure 3—figure supplement 3*), suggesting that our conclusions on the LQTS-like F351A is also relevant for the LQTS mutant F351L.

One of the mutations, F193L, has only minor effects on the biophysical properties of $K_V7.1$ +KCNE1. This mutant was previously reported to have reduced current amplitude compared to the wild-type $K_V7.1$+KCNE1 channel and a mild clinical phenotype (*Yamaguchi et al., 2003*). The F193L mutation may therefore cause loss of function by faster deactivation kinetics and lower current density. Heterozygous expression of mutated subunits and wild-type subunits in equal molar ratios results in general in a milder biophysical phenotype (more close to the wild-type phenotype). This is in line with a milder clinical phenotype generally reported for heterozygous carriers of LQTS mutations compared to individuals with homozygous genotypes (*Priori et al., 1998*; *Jackson et al., 2014*; *Zhang et al., 2008*). Moreover, for different mutations different biophysical effects of the mutations could be dominant or recessive: For S225L and L251A, heterozygous expression in the presence of KCNE1 partially or completely restores wild-type like $V_{50}$, whereas heterozygous expression does not improve closing kinetics compared to homozygous expression. For KCNE1/K70N and KCNE1/S74L, co-expression with wild-type KCNE1 subunits also restores wild-type like $V_{50}$, whereas wild-type like closing kinetics is only partially restored. In contrast, for $K_V7.1$/R583C, heterozygous expression restores wild-type like closing kinetics, but not wild-type like $V_{50}$. However, because of uncertainties regarding the stoichiometry of mutant to wild-type subunits in assembled $K_V7.1$ +KCNE1 channels (as mentioned in the Results section), further studies will be required to understand the mechanisms underlying these apparent dominant or recessive effects and to evaluate possible physiological impact of these effects.

Our results show that all tested mutants respond to N-AT. This is in contrast to previously reported $K_V7$ channel activators on disease-causing $K_V7$ mutants, for which mutants show markedly different sensitivity (*Seebohm et al., 2003*; *Xiong et al., 2007*; *Leitner et al., 2012*). 70 µM N-AT shifts the $G(V)$ curve of the wild-type $K_V7.1$+KCNE1 channel and of all LQTS and LQTS-like mutants by approximately (–50)–(–30) mV, accelerates channel opening and slows down channel closing. In the presence of 70 µM N-AT, the $V_{50}$ of all LQTS and LQTS-like mutants are similar to or more negative than $V_{50}$ for the wild-type $K_V7.1$+KCNE1 channel. For most mutants, 70 µM N-AT overcompensates for the shift in $G(V)$ and reduction in current amplitude caused by the mutations, indicating that a lower N-AT concentration or a less potent N-AT analogue could be used to restore wild-type like $G(V)$ and current amplitudes. Moreover, $K_V7.1$+KCNE1 opening and closing kinetics are partially or completely restored by N-AT. Also, although the disease aetiology of the F193L mutant is likely mainly reduced channel expression, the N-AT induced augmentation caused by a shift in $G(V)$ and increased currents may at least in part overcome the reduction in currents caused by the reduced channel expression. This general ability of N-AT to, at least partly, compensate for the reduced function of mutants with mutations in different parts of the $K_V7.1$+KCNE1 channel complex and with seemingly different molecular defects, as long as a population of these mutant channels reaches the plasma membrane, suggests that N-AT is an interesting model compound for development of future anti-arrhythmics to treat LQTS caused by diverse $K_V7.1$+KCNE1 mutations.

Defective trafficking of mutant $K_V11.1$ ion channels is a common cause of LQTS type 2. About 80-90% of LQTS type 2-associated hERG mutants are estimated to suffer from defective trafficking (*Anderson et al., 2014*; *Sanguinetti, 2010*). The corresponding number for LQTS-associated $K_V7.1$ and KCNE1 mutants is not known. Previous studies identify both trafficking defective and trafficking competent $K_V7.1$ and KCNE1 mutants, e.g. (*Anderson et al., 2014*; *Sanguinetti, 2010*). We are mainly interested in understanding the mechanism that underlies abnormal gating of $K_V7.1$ and KCNE1 mutants. To avoid mutants with severe trafficking defects, we therefore selected mutants that have previously been shown to localize abundantly enough to the cell membrane to generate detectable $K^+$ currents. Several of the selected mutants have been shown to traffic well in mammalian systems ($K_V7.1$/V215M and KCNE1/S74L [*Eldstrom et al., 2010*; *Harmer et al., 2010*]) or generate clearly detectable currents in mammalian cells ($K_V7.1$/R583C [*Yang et al., 2002*]). Our *Xenopus* oocyte experiments that compare mutant current amplitudes with wild-type current amplitudes (*Figure 1—figure supplement 4*) suggest that the reduced ability of the selected mutants to generate

currents in *Xenopus* oocytes may largely be explained by the shifted $G(V)$ of mutants. Trafficking defects could be disguised in *Xenopus* oocytes that are cultured at low temperatures that may rescue some trafficking defects (*Anderson et al., 2014*; *Delisle et al., 2004*). These current amplitude experiments should therefore be interpreted with caution until trafficking of specific $K_V7.1$ and KCNE1 LQTS mutants in mammalian systems has been explored. Previous studies show that membrane expression of trafficking-defect channel mutants (e.g. for $K_V11.1$ and CFTR) can be pharmacologically rescued using compounds that are suggested to stabilize channel conformation during folding and trafficking (*Anderson et al., 2014*; *Delisle et al., 2004*; *Sato et al., 1996*). However, rescue of membrane expression may only partially compensate for mutation-induced loss of function, if these mutants also suffer from defective gating (*Perry et al., 2016*). Our proposed N-AT model for pharmacological correction of '$G(V)$' LQTS mutants could therefore potentially complement pharmacological correction of trafficking-defect LQTS mutants to improve the outcome of patients suffering from LQTS.

We previously suggested that polyunsaturated fatty acids and their analogues (such as N-AT) attract the voltage sensor S4 in $K_V7.1$ by an electrostatic mechanism and thereby shift the $G(V)$ towards more negative voltages and speed up channel opening (*Liin et al., 2015*). We therefore initially hypothesized that N-AT only would restore the function of those LQTS mutations with altered S4 movement. We were pleasantly surprised when N-AT seems to be able to restore the function of many LQTS and LQTS-like mutants, with diverse mutational defects (such as S225L and F351A). Using voltage clamp fluorometry, we have previously shown that both the main gating charge movement and the gate opening of $K_V7.1$+KCNE1 channels are accompanied by fluorescence signals from fluorophores attached to S4 (*Barro-Soria et al., 2014*). This suggests that S4 moves both during the main gating charge movement and during the subsequent channel opening in $K_V7.1$+KCNE1 channels (*Barro-Soria et al., 2014*), which is similar to observations in Shaker $K_V$ channels (*Börjesson and Elinder, 2011*; *Pathak et al., 2005*; *Phillips and Swartz, 2010*). Therefore, N-AT could affect both the main gating charge movement and gate opening by acting on the S4 voltage sensor, as has been shown for hanatoxin which targets the voltage-sensing domain in the Shaker $K_V$ channel (*Milescu et al., 2013*). This hypothesis is supported by our voltage-clamp fluorometry experiments using $K_V7.1$/S225L and $K_V7.1$/F351A in which N-AT accelerates the fluorescence components associated with both the main S4 movement ($F1$) and gate opening ($F2$), as well as accelerates the kinetics of channel opening. This proposed mechanism would explain why N-AT can restore the function of mutations that mainly target either the main S4 movement or gate opening. However, the dramatic decrease in the fluorescence signal caused by N-AT makes it hard for us to completely determine the effect of N-AT on the $F(V)$ of mutants. Therefore, the complete mechanism of N-AT in the different mutations is not clear.

Future studies are required to assess the clinical utility of PUFA analogues in cardiomyocytes and animal models. We see channel specificity of PUFA analogues as one major challenge and recognize the need to improve PUFA analogue affinity to $K_V7.1$+KCNE1 to reduce required therapeutic concentrations and minimize potential adverse effects. Despite these challenges, our data show that the magnitude of the N-AT-induced voltage shifts are in a similar range as the shifts induced by several LQTS mutations, thereby serving as proof of concept that this PUFA analogue, at least partly, restores channel function in diverse LQTS and LQTS-like mutants.

## Materials and methods

Experiments were approved by The Linköping Animal Ethics Committee at Linköping University and The Animal Experiments Inspectorate under the Danish Ministry of Food, Agriculture and Fisheries (University of Copenhagen).

### Experiments on *Xenopus* laevis oocytes

#### Molecular biology

Expression plasmids human $K_V7.1$ (GenBank Acc.No. NM_000218) in pXOOM and KCNE1 (NM_000219) in pGEM have been previously described (*Jespersen et al., 2002*; *Schmitt et al., 2007*). LQTS and LQTS-like point mutations and G219C were introduced into $K_V7.1$ or KCNE1 using site-directed mutagenesis (QuikChange Stratagene, CA). All newly generated constructs were sequenced to ensure integrity (Genewiz, NJ). cRNA was prepared from linearized DNA using the T7

mMessage mMachine transcription kit (Ambion, TX). RNA quality was checked by gel electrophoresis, and RNA concentrations were quantified by UV spectroscopy.

## Two-electrode voltage-clamp electrophysiology

*Xenopus* laevis oocytes (from EcoCyte Bioscience, TX, or prepared in house) were isolated and maintained as previously described (*Börjesson et al., 2010*). 50 nl cRNA (~50 ng $K_V7.1$ for $K_V7.1$-only expression, 25 ng $K_V7.1$ + 8 ng KCNE1 for homozygous expression, or 12.5 ng $K_V7.1^{wt}$ + 12.5 ng $K_V7.1^{mut}$ + 8 ng KCNE1$^{wt}$ alternatively 25 ng $K_V7.1^{wt}$ + 4 ng KCNE1$^{wt}$ + 4 ng KCNE1$^{mut}$ for heterozygous expression) was injected into each oocyte. Currents were measured at room temperature 2–5 days after injection with the two-electrode voltage-clamp technique (CA-1B amplifier, Dagan, MN). For the current amplitude experiments presented in *Figure 1—figure supplement 4*, the current amplitude of mutants were normalized to the current amplitude of wild-type $K_V7.1$+KCNE1 expressed in the same batch of oocytes and incubated under identical conditions for the same time period. Currents were sampled at 1–3.3 kHz, filtered at 500 Hz, and not leakage corrected. The control solution contained (in mM): 88 NaCl, 1 KCl, 15 HEPES, 0.4 $CaCl_2$, and 0.8 $MgCl_2$ (pH adjusted to 7.4 using NaOH). The holding voltage was generally set to –80 mV. Activation curves were generally elicited by stepping to test voltages between –110 and +60 mV (3–5 s durations and 10 mV increments) followed by a tail voltage of –20 mV. Voltage clamp fluorometry experiments were performed as previously described on oocytes labeled for 30 min with 100 µM Alexa-488-maleimide (Molecular Probes) at 4°C (*Barro-Soria et al., 2014*; *Osteen et al., 2010*; *Osteen et al., 2012*). For voltage clamp fluorometry experiments on $K_V7.1$/G219C*, the holding voltage was –80 mV, the pre-pulse –120 mV for 2 s, and test voltages ranging between –140 and +80 mV for 3 s in 20 mV increments. The tail voltage was –80 mV. For $K_V7.1$/G219C*/KCNE1, the holding voltage was –80 mV, the pre-pulse –160 mV for 5 s, and test voltages ranging between –160 and +80 mV for 5 s in 20 mV increments. The tail voltage was –40 mV. N-arachidonoyl taurine was purchased from Cayman Chemical (MI, USA) and stored, diluted and applied to the oocyte chamber as previously described (*Liin et al., 2015*). Control solution was added to the bath using a gravity-driven perfusion system.

## Electrophysiological analysis

To quantify effects on the *G(V)*, tail currents (measured shortly after initiation of tail voltage) were plotted against the pre-pulse (test) voltage. The following Boltzmann relation was fitted to the data

$$G_K(V) = G_{max}/(1 + exp((V_{50} - V)/s)), \tag{1}$$

where $V_{50}$ is the midpoint (i.e. the voltage at which the conductance is half the maximal conductance estimated from the fit) and s the slope factor (shared slope for control and N-AT curves within the same cell). In figures showing $I_{tail}$ vs voltage, the curves are normalized to the fitted $G_{max}$. The same single Boltzmann relation was used to fit the *F(V)* from voltage clamp fluorometry recordings of $K_V7.1$ without KCNE1 co-expression, where fluorescence at the end of the test pulse was plotted versus the test voltage (*Barro-Soria et al., 2014*). For voltage-clamp fluorometry recordings of $K_V7.1$ with KCNE1 co-expression (and F351A without KCNE1), a double Boltzmann relation was used (*Barro-Soria et al., 2014*). For experiments where conductance or fluorescence did not clearly show signs of saturation in the experimental voltage range, these fits should be considered as an approximation. To estimate the effect of N-AT on Gibbs free energy, the following relation was used:

$$\Delta\Delta G_o = z * \Delta V_{50} * F, \tag{2}$$

Where z is the gating charge of each channel deduced from the slope of the Boltzmann fits according to $z = 25/s$, $\Delta V_{50}$ is the N-AT induced shift in the $V_{50}$ values from the Boltzmann fits, and F is Faraday's constant (*Li-Smerin and Swartz, 2001*; *Monks et al., 1999*; *DeCaen et al., 2008*). This analysis assumes a two-state model and tends to underestimate the z (*Chowdhury and Chanda, 2012*). The calculated $\Delta\Delta G_o$ should therefore be seen as an approximation. For opening and closing kinetics, $T_{50,open}$ was defined as the time it takes to reach 50% of the current in the end of a 3 s (5 s for KCNE1 co-expression) long test pulse to +40 mV. $T_{50,close}$ was defined as the time it takes to reduce the amplitude (= instantaneous tail current – steady state tail current) of the tail current by 50% when stepping to a tail pulse to –20 for 5 s. To analyze the effect of N-AT on fluorescence and

current kinetics, single or double exponentials were fitted to the fluorescence or current traces. The ratios of time constants before and after application of N-AT were then calculated.

## Modeling

Fluorescence and currents from the $K_V7.1+KCNE1$ models were simulated using Berkeley Madonna (Berkeley, CA).

## Statistics

Average values are expressed as mean ± SEM. Mutant parameters (e.g. $V_{50}$ and $\Delta\Delta G_o$) were compared to wild-type parameters using one-way ANOVA with Dunnett's Multiple Comparison Test. Comparison of homozygous and heterozygous expression was done using one-way ANOVA with pair-wise Bonferroni's Test. The effects of N-AT on fluorescence and current kinetics were analysed using two-tailed one sample t-test where ratios were compared with a hypothetical value of 1. $p < 0.05$ is considered as statistically significant.

## Acknowledgements

We thank Frida Starck Härlin (Linköping University) and Briana Watkins (University of Miami) for help with some experiments and Drs. Fredrik Elinder (Linköping University), Laura Bianchi and Feng Qiu (University of Miami), Nicole Schmitt, Mark Skarsfeldt and Federico Denti (University of Copenhagen) for valuable comments.

## Additional information

### Competing interests

SIL, HPL: A patent application (62/032,739) based on these results has been submitted by the University of Miami with SIL and HPL identified as inventors. The other authors declare that no competing interests exist.

### Funding

| Funder | Grant reference number | Author |
|---|---|---|
| National Institutes of Health | R01GM109762 | H Peter Larson |
| American Heart Association | 16GRNT30990060 | H Peter Larson |
| Svenska Sällskapet för Medicinsk Forskning | | Sara I Liin |
| Vetenskapsrådet | 524-2011-6806 | Sara I Liin |
| Northwest Lions Foundation | | Sara I Liin |

The funders had no role in study design, data collection and interpretation, or the decision to submit the work for publication.

### Author contributions

SIL, JEL, Conception and design, Acquisition of data, Analysis and interpretation of data, Drafting or revising the article; RB-S, Acquisition of data, Analysis and interpretation of data, Drafting or revising the article; BHB, Conception and design, Drafting or revising the article; HPL, Conception and design, Analysis and interpretation of data, Drafting or revising the article

### Author ORCIDs

Sara I Liin, http://orcid.org/0000-0001-8493-0114

### Ethics

Animal experimentation: Experiments were performed in strict accordance with the recommendations of The Linköping Animal Ethics Committee at Linköping University and The Animal Experiments Inspectorate under the Danish Ministry of Food, Agriculture and Fisheries. Protocols were approved by The Linköping Animal Ethics Committee at Linköping University (#53-13 ) and The

Animal Experiments Inspectorate under the Danish Ministry of Food, Agriculture and Fisheries (University of Copenhagen; #2014-15-2934-01061).

## Additional files

**Supplementary files**

• Supplementary file 1. Summary of biophysical properties of LQTS and LQTS-like mutations for single injection (left panel) and homozygous co-injection (right panel).

• Supplementary file 2. Summary of biophysical properties of LQTS and LQTS-like mutations for homozygous (left panel) and heterozygous expression (right panel).

• Supplementary file 3. Summary of effect of 70µM N-AT on WT and LQTS and LQTS-like mutants for homozygous (left panel) and heterozygous.

• Supplementary file 4. Parameters for KV7.1 model (a) andKV7.1+KCNE1 model (b)in *Figure 3—figure supplement 1*.

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
