## [Decision Letter]

Thank you for submitting your article "Fatty Acid Analogue N-Arachidonoyl Taurine Restores Function of I_Ks_ Channels with Diverse Long QT Mutations" for consideration by *eLife*. Your article has been favorably evaluated by Richard Aldrich (Senior Editor) and three reviewers, one of whom, Kenton J Swartz (Reviewer #1), is a member of our Board of Reviewing Editors. The following individual involved in review of your submission has agreed to reveal his identity: Christopher J Lingle (Reviewer #2).

The reviewers have discussed the reviews with one another and the Reviewing Editor has drafted this decision to help you prepare a revised submission.

This paper shows that LQTS deficits resulting from diverse loss-of-function mutations in K_v_7.1+KCNE1 channels can potentially be rescued by the polyunsaturated fatty acid analogue, N-arachidonyl taurine (N-AT). To make this point, the authors focus on two mutations, S225L and F351A, drawing the conclusion that the S225L mutation produces a rightward shift in voltage-sensor activation while the F351A mutation largely results in less effective channel opening once voltage-sensors are activated. The results are nicely supported by a comparison of fluorescence-voltage and conductance-voltage curves for each construct with for K_v_7.1 expressed alone or when coexpressed with KCNE1. To validate the idea that each mutation does reduce activation in mechanistically distinct ways, the authors use previously developed models of K_v_7.1 activation +/- KCNE1 to show that a 50 mV rightward shift in VSD activation nicely accounts for both the FV and GV shifts for the S225L mutation, while the F351A mutations can be approximated by a +140 mV shift in the voltage-dependence of channel opening. Although this approach to validate the idea that the two mutations inhibit K_v_7.1 activation by distinct mechanism might be viewed as somewhat dependent on the validity of the activation models (which remains a bit uncertain), the fact that changes in largely a single parameter accounts for the FV and GV relationships both in the absence and presence of KCNE1 is satisfying.

The authors make the argument that, with appropriate development of compounds of greater specificity and affinity, this may be a plausible strategy for treatment of various LQTS diseases. A number of arguments can be made against the idea that this approach could ever work, such as:

1) Deficits in expression may underlie the effects of many LQTS mutations;

2) Fatty acid analogues are likely to have effects on many targets;

3) Concentrations of any potentially effective compound may have to be adjusted for any given LQTS mutation.

However, overall, we felt that the authors did a good job of discussing many of these challenges and providing perspective about what would have to be done before this approach might be of benefit. As a proof of principle that an activator of K_v_7.1/KCNE channels can rescue function, irrespective of the nature of the functional deficit, this paper accomplishes that goal. The idea that an activator may qualitatively restore a channel's function irrespective of the origins of the original gating deficit is intriguing.

Essential revisions:

More data needs to be included to show the effects of the PUFA on fluorescence signals. The text states that the compound greatly diminishes fluorescence but otherwise little is shown or explained. Do the authors understand the mechanism? Is N-AT a collisional quencher? Is there any voltage-dependence to quenching? If not, why not show F-V relations unscaled and then scaled so the reader can appreciate what is going on? Given the large effects of N-AT on fluorescence, should we be concerned about interpreting small changes in kinetics? Were the traces in Figure 5 normalized to allow comparison of the temporal properties? Although the authors would like to conclude that N-AT has generalized effects that affect both the VSD activation and channel opening, it seems like it could still be the case that effects on channel opening are primary. As such, the FV and GV data might help put this issue on firmer foundation.

[Editors’ note: a previous version of this study was rejected after peer review, but the authors submitted for reconsideration. The first decision letter after peer review is shown below.]

Thank you for submitting your work entitled "Fatty Acid Analogue N-Arachidonoyl Taurine Restores Function of I_Ks_ Channels with Diverse Long QT Mutations" for consideration by *eLife*. Your article has been favorably evaluated by Richard Aldrich (Senior Editor) and three reviewers, one of whom, Kenton J Swartz (Reviewer #1), is a member of our Board of Reviewing Editors. The following individual involved in review of your submission has agreed to reveal his identity: Christopher J Lingle (Reviewer #3).

Our decision has been reached after consultation between the reviewers. Based on these discussions and the individual reviews below, we regret to inform you that your work will not be considered further for publication in *eLife*.

All three reviewers found your work to be interesting because it attempts to understand disease causing mutations at a mechanistic level and provides proof of concept evidence that PFU could potentially be used to correct the deficit. Although the reviewers were somewhat divergent in their initial enthusiasm for the manuscript, after discussion there was consensus that the study would need substantial work to be appropriate for *eLife*. All three reviewers offer specific suggestions, which you should consider carefully in going forward with publication. These include 1) clarification of the mechanism by which the two studied mutants affect gating, 2) examination of the impact of mutations on surface levels of the KCNQ channel in order to assess whether restoring the gating properties with PUFs would be sufficient to correct the deficits produced by the mutants, and 3) clarification of the impact of PUFs on other ion channels, cardiac action potentials and EKGs. The overarching concern was that insufficient evidence was presented to support the potential therapeutic utility of PUFs. Our policy is to reject manuscripts requiring the amount of effort that we feel yours would require, but we hope that the reviews will be helpful in revising your manuscript for publication elsewhere. If you feel that you can address the concerns of the reviewers, we would be willing to reconsider the manuscript as a new submission.

*Reviewer #1:*

The authors investigate a series of mutations in the KCNQ channel that cause long QT syndrome in humans and use voltage-clamp fluorimetry with a fluorescent probes attached to the voltage sensors to track conformational changes while measuring channel activation. They see that the mutants alter the voltage-activation relations to varying degrees, but typically shifting activation to positive voltages, slowing activation and speeding deactivation. From comparisons of the F-V and G-V relations in the absence and presence of KCNE, the authors propose that the one mutant located in S4 alters voltage-sensor activation while another located in S6 alters pore opening. They then show that a polyunsaturated fatty (PFU) acid can correct the effects of the mutants, and do proof of concept experiments in iPSC-derived cardiomyocytes and guinea pig hears where a KCNQ inhibitor is used to induce LQT and show that the PUF can restore normal LQT. This is an interesting study because it attempts to understand disease causing mutations at a mechanistic level and provides proof of concept evidence that PFU could potentially be used to correct the deficit. Although the authors' interpretations are plausible, the data are not always so clean and I can imagine more complex mechanisms may be involved.

1) The discussion of the combined F-V and G-V data in Figure 3 in the subsection “K_V_7.1+KCNE1 channel dysfunction is caused by different underlying mechanisms” needs some work and it is not clear to me that S225L affects voltage-sensor activation and F351A affects pore opening. The data to me seem much more complex than the authors acknowledge. I would like to see a clearer presentation of the evidence supporting the key conclusions and a more nuanced discussion of the underlying complexity. The authors may be correct, but I am not sure we understand why the F-V and G-V overlap for KCNQ alone and what KCNE is doing to those relations, which makes interpretation of what is seen with the mutants less than straightforward.

2) What is the logic of using N-AT instead of other PUFs the authors have previously studied? Given those studies, I would imagine N-AT might have substantial effects on other K_v_ channels, and possibly even Nav and Cav channels. What makes the authors think that N-AT will have any selectivity for KCNQ? This feels like a stretch to imagine using these in humans at high μM concentrations.

*Reviewer #2:*

This paper investigates a selection of Long QT syndrome Type 1 and Type 5 mutations that shift the voltage-dependence of channel opening. Like others, they have found that such mutants have altered voltage-dependence of channel opening, and changed closing kinetics, that can be partially normalized by "heterozygous" expression. N-AT moves activation back to more negative potentials in the LQT mutants, as the authors have previously shown for I_Ks_, and N-AT accelerates all components of fluorescence movement from the mutant channels incorporating G219C. In addition, it is shown that N-AT can affect action potential duration in iPSC-derived cardiomyocytes and intact guinea-pig hearts, although these data are less convincing. The significance of this work is that N-AT is an activator of both wild type K_v_7.1 +KCNE1 and LQTS mutants. It does not seem so surprising, or indeed particularly significant that N-AT affects LQT mutants in the same way as WT channels. The authors have already reported that PUFA analogues shift the activation gating of K_v_7.1+KCNE1 channels, so it is to be expected that the LQT mutants have their activation affected as well, as simple missense mutants. Given that, most of the biophysics presented is logical and seems to suggest that N-AT acts non-specifically on the channels (unlike their previous suggestion that WT channels are affected by an "electrostatic mechanism" affecting the VSD). For most in the field, interest in this work is lessened as it is generally accepted that about 2/3 of LQT phenotypes come about as a result of lowered expression at the cell surface from misfolding and other events during synthesis and trafficking.

1) Expression in oocytes bypasses these problems due to the low incubation temperatures, so it is unclear whether these gating effects are relevant in the greater scheme of increasing overall current density at the mammalian cell surface. Do the authors know what the expression and biophysical characteristics are for these LQT mutants at physiological temperature compared with wild type to understand how N-AT may alter the abnormal functioning of these mutants?

2) There is a little bit of mixture of models in this study. Most experiments are carried out at room temperature in oocytes, but sometimes CHO cells are used for hKv11.1, derived cardiomyocytes and intact hearts are used for action potential and QT studies. Some data that could tie together studies in the different models would support the potential translational importance of the idea.

3) The data from iPSC-derived cardiomyocytes and intact guinea-pig hearts are unconvincing. In the intact hearts it looks like other changes affect the early action potential to shorten it in the presence of N-AT. Changes in the isoelectric region of the EKG suggest a dispersion of action potential effects by N-AT. In intact hearts, the slope of phase 3 repolarization in the HMR+N-AT trace is shallower than in control or with HMR1556 alone. This suggests less net outward current when N-AT is added, the opposite of what is contended. Perhaps this is because N-AT blocks many other cardiac currents.

4) The authors suggest that PUFAs could provide a treatment option for LQTS, but there is a large literature on the effects of PUFAs on humans, in animal models and also on individual ion currents in heart showing that almost all ion currents, inward and outward, including I_Kr_, are decreased by PUFAs. The well-established exception is I_Ks_, but the overall effect on the QT interval duration could be pro- or antiarrhythmic depending on the physiology of the tissue and the pathology affecting the heart.

*Reviewer #3:*

The main point of this paper is that LQTS deficits regulating from diverse loss-of-function mutations in K_v_7.1+KCNME1 channels can potentially be rescued by the polyunsaturated fatty acid analogue, N-arachidonyl taurine (N-AT). This is an exciting and interesting conclusion. The paper is clearly written and the results are straight forwards and support the conclusions. A particularly compelling aspect of the paper is that the authors show that N-AT can rescue function in two mechanistically distinct types of LQTS mutants, those likely to affect voltage sensor function and those affecting channel opening equilibria. This provides some assurance that this sort of pharmacological strategy might have general applicability for a number of different categories of LQTS mutations.

1) One topic that probably requires some additional clarification concerns the presentation of GVs and how the *V*50's and z values were generated. For many of the mutant GV curves, a saturating level of activation is never obtained, but the plotted GV curves are simply normalized to the maximum level of conductance. I think for some (many) readers this may give a misleading impression of what the relative conductances as a function of voltage may be among different constructs (Figure 1, Figure 2, and others) or in response to N-AT application (e.g., Figure 4), since the assumption may be made that the voltage of the half maximal conductance after normalization corresponds to the true *V*50. Although sometimes it is natural to not trust the extrapolated gmax from a Boltzmann fit, plotting the GVs normalized to the fitted Gmax may be a better proxy for the true channel behavior. In such cases, the observed 0.5 level of conductance directly informs a reader of the approximate Vh, and it also informs the reader regarding changes in the slope, which may otherwise be obscured. In Figure 4, the normalization used by the authors probably tends to minimize the effects at the lower concentrations, while normalizing to the fitted gmax might avoid this. One might also ask in regard to Figure 4, since the control and N-AT tails are all measured in the same patches, why not use the maximal conductance with 70 μM N-AT as the basis for normalization? My concern regarding the GV displays and the estimates of Vh and z become most critical in regards to the DDG estimates (I think, as the process is described in the methods, this was done appropriately, but some clarifications about this might help).

---

## [Author Response]

*Essential revisions:*

*More data needs to be included to show the effects of the PUFA on fluorescence signals. The text states that the compound greatly diminishes fluorescence but otherwise little is shown or explained. Do the authors understand the mechanism? Is N-AT a collisional quencher? Is there any voltage-dependence to quenching? If not, why not show F-V relations unscaled and then scaled so the reader can appreciate what is going on? Given the large effects of N-AT on fluorescence, should we be concerned about interpreting small changes in kinetics? Were the traces in Figure 5 normalized to allow comparison of the temporal properties? Although the authors would like to conclude that N-AT has generalized effects that affect both the VSD activation and channel opening, it seems like it could still be the case that effects on channel opening are primary. As such, the FV and GV data might help put this issue on firmer foundation.*

We agree that the previous description of N-AT-induced reduction in overall fluorescence intensity was minimal. To test whether N-AT is a collisional quencher, we monitored the fluorescence from unbound Alexa488 in control solution and in N-AT-supplemented control solution (up to 0.5 M N-AT). In these experiments where no oocytes or channels were present, we did not see any quenching effect of N-AT on Alexa488 fluorescence, suggesting that N-AT is not a collisional quencher of Alexa488. N-AT actually slightly increases the Alexa488 fluorescence by 10-20% (in 0.5 M N-AT, Figure 5—figure supplement 1). Although further experiments will be required to fully understand how N-AT decreases the fluorescence from Alexa-488 bound to K_v_7.1+KCNE1 channels, these experiments suggest that the reduced fluorescence observed in our voltage clamp fluorometry experiments are not caused by direct N-AT quenching of the fluorophore. Instead, it is possible that N-AT induces some rearrangement in the channel that brings some unknown K_v_7.1 or KCNE1 residue, such as a trp, close to Alexa488 and thereby N-AT indirectly quenches the fluorescence.

Due to the slow kinetics of the K_v_7.1+KCNE1 channel and the wide voltage range over which it activates, it takes about 7 minutes to record a full *F*(*V*) curve on K_v_7.1+KCNE1. During this time course, roughly 50% of the fluorescence signal disappears (now shown in Figure 5—figure supplement 1). As a consequence, the voltage-dependent changes in fluorescence intensity caused by the S4 movement will be underestimated at more positive voltages compared to negative voltages (we run the protocol from negative to positive test voltages). In addition, the *F*(*V*) curves for WT K_v_7.1+KCNE1 and K_v_7.1/F351A+KCNE1 channels are hard to normalize (especially when N-AT has reduced the fluorescence signal to noise), due to not complete saturation at either the negative or positive end of the *F*(*V*) curve. We therefore feel that we presently are unable to reliably quantify N-AT-induced shifts in the *F*(*V*) curve for WT K_v_7.1+KCNE1 and K_v_7.1/F351A+KCNE1. However, we now include a normalized *F*(*V*) curve of K_v_7.1/G219C*/S225L+KCNE1 (which shows better saturation at both ends of the *F*(*V*) curve) in the presence of N-AT, to compare to the corresponding *F*(*V*) in control solution (Figure 5—figure supplement 1). This normalized *F*(*V*) curve indicates that N-AT shifts both the first part of the *F*(*V*) curve (which represents the main outward S4 movement, i.e. VSD activation) and the second part of the *F*(*V*) curve (which represents S4 rearrangements associated with channel opening) towards more negative voltages.

We now clarify in the figure legend that fluorescence traces in Figure 5 are normalized to allow kinetic comparison.

To fully understand the mechanism by which N-AT and other PUFA analogues activate the K_v_7.1+KCNE1 channel, we need to develop means to reliably measure complete *F*(*V*) curves in future studies.

[Editors’ note: the author responses to the first round of peer review follow.]

*Reviewer #1:*

*1) The discussion of the combined F-V and G-V data in Figure 3 in the subsection “K_V_7.1+KCNE1 channel dysfunction is caused by different underlying mechanisms” needs some work and it is not clear to me that S225L affects voltage-sensor activation and F351A affects pore opening. The data to me seem much more complex than the authors acknowledge. I would like to see a clearer presentation of the evidence supporting the key conclusions and a more nuanced discussion of the underlying complexity. The authors may be correct, but I am not sure we understand why the F-V and G-V overlap for KCNQ alone and what KCNE is doing to those relations, which makes interpretation of what is seen with the mutants less than straightforward.*

We agree that the previous discussion of the mechanism of the two mutations were minimal. We have now included some more discussion and added kinetic modeling to show that our proposed mechanisms for these two mutations are plausible. In this kinetic modeling, we have used the two models that our group has previously developed for wild type K_v_7.1 and I_Ks_ channels (Osteen et al., 2012; Barro-Soria et al., 2014). These models are based on extensive data from many types of experiments (ionic currents, gating currents, fluorescence, linked constructs, complex voltage protocols). We feel that we have earlier extensively shown that the main reason for why the GV and FV overlap are that K_v_7.1 channels open after only a subset of S4s are activated (Osteen et al., 2012). In contrast, I_Ks_ channels need all four S4s to be activated before they open (Osteen et al., 2012; Barro-Soria et al., 2014). The main result supporting these two activation mechanisms is our data from linked K_v_7.1 subunits with different numbers of permanently activated S4s (Osteen et al. 2012). Linked K_v_7.1 channels with two permanently activated S4s are open even at hyperpolarized potentials where the two wt S4s have clearly deactivated, showing that K_v_7.1 channels open even with a subset of S4s activated. In contrast, the same linked construct co-expressed with KCNE1 is closed at hyperpolarized voltages, showing that I_Ks_ channels don’t open after only a subset of S4s have activated. Using these two models, we can show that S225L can be explained in both models by just shifting the voltage dependence of the main S4 movement, whereas F351A can be explained in both models by just shifting the voltage dependence of opening. That the effects of each mutation can be explained in both the K_v_7.1 and I_Ks_ models by changing a single parameter (different in each mutation) by the same amount is a strong argument for the robustness of the models and the proposed hypothesized mechanisms of the two mutations. Of course, more complex mechanisms of action of these mutations could also generate the same effects on the currents and fluorescence, but we here present the mechanisms we feel are the most simple that can explain most of the phenotypes.

Other models for K_v_7.1 and I_Ks_ have been proposed, but none of these have been as extensively tested as ours and, in reality, these models fail some of our previous experimental tests. The model that is most similar to ours is the model from Jianmin Cui’s group (Zaydman et al., 2014). In principle, Jianmin’s and our models are very similar: the main difference is that their models have more states and ours are simplified versions of their models. Unfortunately, they use parameters from the model by Rudy and Silva (e.g. a slow S4 and a fast gate), which we have shown is not compatible with more complex triple pulse voltage protocols (Barro-Soria et al., 2014). So the actual parameters in their model would have to be refitted to more complex voltage protocols before their models can be used. However, we now acknowledge that there are other models for K_v_7.1 and I_Ks_ and that our conclusions about the mechanism of the mutations are model dependent. However, independent of what models are used to describe the channel, it is clear from the differences in their effects on the fluorescence that these two mutations affect the channels by different mechanisms and that N-AT restores most of the function in both mutations.

*2) What is the logic of using N-AT instead of other PUFs the authors have previously studied? Given those studies, I would imagine N-AT might have substantial effects on other K_v_ channels, and possibly even Nav and Cav channels. What makes the authors think that N-AT will have any selectivity for KCNQ? This feels like a stretch to imagine using these in humans at high μM concentrations.*

We have toned down the clinical utility of N-AT and instead termed it as a good starting compound for future drug development. We use N-AT because it gives large robust effects. However, its affinity is not ideal. We have recently been able to combine, in novel PUFA compounds, the robust effects of N-AT with the higher affinity effects of other PUFAs. These compounds need further testing and will be presented in future work.

*Reviewer #2:*

*1) Expression in oocytes bypasses these problems due to the low incubation temperatures, so it is unclear whether these gating effects are relevant in the greater scheme of increasing overall current density at the mammalian cell surface. Do the authors know what the expression and biophysical characteristics are for these LQT mutants at physiological temperature compared with wild type to understand how N-AT may alter the abnormal functioning of these mutants?*

We agree that trafficking could be a problem for these mutations. However, we do show that for all of the mutations there are gating defects that would be in addition to any putative trafficking effects. Our goal with N-AT is to restore the gating defects, not to restore any trafficking defects per se. We show in *Xenopus oocytes* that N-AT can restore the function of all mutants, as measured by the current amplitude of wt channels in control solutions compared to mutant channels in N-AT solutions (Figure 4, Figure 4—figure supplement 4). In addition, in *Xenopus oocytes*, most reductions in current amplitude (if not all) can be explained by the gating defects caused by the mutations (Figure 1—figure supplement 4). So, in *Xenopus oocytes*, we don’t see much of any trafficking defect in these mutants.

However, as the reviewer points out, there might be trafficking defects in mammalian cells that are not present in *Xenopus oocytes*. Other studies on trafficking and mammalian cells are now referred to (Eldstrom et al., 2010; Harmer et al., 2010). For any mutation that displays trafficking defects, one has to find a cure for this trafficking defect. However, this is outside of the scope of our study. But even if the trafficking is corrected, these mutants would also need to be corrected for their gating defect. Otherwise, one would end up with surface-expressed, defective I_Ks_ channels, which could still cause arrhythmia. We now clearly state all these caveats and what N-AT could do to these mutants in these possible cases.

*2) There is a little bit of mixture of models in this study. Most experiments are carried out at room temperature in oocytes, but sometimes CHO cells are used for hKv11.1, derived cardiomyocytes and intact hearts are used for action potential and QT studies. Some data that could tie together studies in the different models would support the potential translational importance of the idea.*

The data from CHO, cardiomyocytes, and intact hearts have been removed.

*3) The data from iPSC-derived cardiomyocytes and intact guinea-pig hearts are unconvincing. In the intact hearts it looks like other changes affect the early action potential to shorten it in the presence of N-AT. Changes in the isoelectric region of the EKG suggest a dispersion of action potential effects by N-AT. In intact hearts, the slope of phase 3 repolarization in the HMR+N-AT trace is shallower than in control or with HMR1556 alone. This suggests less net outward current when N-AT is added, the opposite of what is contended. Perhaps this is because N-AT blocks many other cardiac currents.*

This data has been removed.

*4) The authors suggest that PUFAs could provide a treatment option for LQTS, but there is a large literature on the effects of PUFAs on humans, in animal models and also on individual ion currents in heart showing that almost all ion currents, inward and outward, including I_Kr_, are decreased by PUFAs. The well-established exception is I_Ks_, but the overall effect on the QT interval duration could be pro- or antiarrhythmic depending on the physiology of the tissue and the pathology affecting the heart.*

We have removed the QT interval data on cardiomyocytes and intact heart. We have toned down the therapeutic effect of N-AT and instead focused on the biophysical effect of PUFA analogues on the I_Ks_ channel.

*Reviewer #3:*

*1) One topic that probably requires some additional clarification concerns the presentation of GVs and how the* V*50's and z values were generated. For many of the mutant GV curves, a saturating level of activation is never obtained, but the plotted GV curves are simply normalized to the maximum level of conductance. I think for some (many) readers this may give a misleading impression of what the relative conductances as a function of voltage may be among different constructs (Figure 1,Figure 2, and others) or in response to N-AT application (e.g., Figure 4), since the assumption may be made that the voltage of the half maximal conductance after normalization corresponds to the true V50. Although sometimes it is natural to not trust the extrapolated gmax from a Boltzmann fit, plotting the GVs normalized to the fitted Gmax may be a better proxy for the true channel behavior. In such cases, the observed 0.5 level of conductance directly informs a reader of the approximate Vh, and it also informs the reader regarding changes in the slope, which may otherwise be obscured. In Figure 4, the normalization used by the authors probably tends to minimize the effects at the lower concentrations, while normalizing to the fitted gmax might avoid this. One might also ask in regard to Figure 4, since the control and N-AT tails are all measured in the same patches, why not use the maximal conductance with 70 μM N-AT as the basis for normalization? My concern regarding the GV displays and the estimates of Vh and z become most critical in regards to the DDG estimates (I think, as the process is described in the methods, this was done appropriately, but some clarifications about this might help).*

We have changed the normalization in all graphs. We also better explain in text how we estimate the *V*50. The *V*50 is always determined from the fit, as suggested by the reviewer.